# A Hierarchy Byzantine Fault Tolerance Consensus Protocol Based on Node Reputation

**DOI:** 10.3390/s22155887

**Published:** 2022-08-06

**Authors:** Xixi Wang, Yepeng Guan

**Affiliations:** 1School of Communication and Information Engineering, Shanghai University, Shanghai 200444, China; 2Key Laboratory of Advanced Display and System Application, Ministry of Education, Shanghai 200072, China

**Keywords:** consensus protocol, reputation model, hierarchy structure, random selection mechanism

## Abstract

A blockchain has been applied in many areas, such as cryptocurrency, smart cities and digital finance. The consensus protocol is the core part of the blockchain network, which addresses the problem of transaction consistency among the involved participants. However, the scalability, efficiency and security of the consensus protocol are greatly restricted with the increasing number of nodes. A Hierarchy Byzantine Fault Tolerance consensus protocol (HBFT) based on node reputation has been proposed. The two-layer hierarchy structure is designed to improve the scalability by assigning nodes to different layers. Each node only needs to exchange messages within its group, which deducts the communication complexity between nodes. Specifically, a reputation model is proposed to distinguish normal nodes from malicious ones by a punish and reward mechanism. It is applied to ensure that the malicious node merely existing in the bottom layer and the communication complexity in the high layer can be further lowered. Finally, a random selection mechanism is applied in the selection of the leader node. The mechanism can ensure the security of the blockchain network with the characteristics of unpredictability and randomicity. Some experimental results demonstrated that the proposed consensus protocol has excellent performance in comparison to some state-of-the-art models.

## 1. Introduction

A blockchain is one of the latest trends in distributed networks, where each node maintains an append-only ledger [1]. It has been applied to solve the trust challenge for large-scale collaborative works with the characteristics of decentralization, non-tampering and traceability [2,3,4]. According to the degree of decentralization, a blockchain is typically divided into three types, including public chain, consortium chain and private chain. The public chain [5] is a network that any node is allowed to participate in at any time. It is usually used in scenarios where a high latency and untrusted nodes are accepted. The consortium chain [6] is shared and managed by several institutions. In the private chain [7], the nodes are all controlled by one institution.

The consensus protocol plays a vital role in the blockchain for ordering transactions and guaranteeing the consistency of the ledger stored in the node. The consensus protocol [8] determines the performance of the blockchain system to a large extent, such as latency, transaction throughput, scalability and so on [9]. Proof-based consensus protocols are widely applied to many public blockchains, such as Proof-of-Work (PoW) [10] in the Bitcoin system, Proof-of-Stake (PoS) [11] and Delegated Proof-of Stake (DPoS) [12] in Ethereum. Such protocols are designed with an excellent node scalability through node competition. However, they are greatly energy-consuming and have a long transaction confirmation delay. For instance, [13] points out that the transaction confirmation delay is typically limited to 10 min in Bitcoin and 3 min in Ethereum, while in a consortium and private blockchain, lighter consensus protocols such as PBFT [14], Paxos [15] and Raft [16] are preferably adopted. These protocols are energy-saving and can achieve a higher throughput.

Practical Byzantine Fault Tolerance (PBFT) is proposed in [14]. It has shown great potential to break the performance bottleneck of the proof-based consensus protocol. PBFT [14] is widely adopted in private and consortium blockchains, as it does not consume much energy and can achieve a much higher throughput. However, the node scalability, which reflects the capacity of the network to process the node growth, is a bottleneck for PBFT [14]. It can only scale to a few tens of nodes due to the high communication complexity [17]. When the number of nodes in the network exceeds this threshold, the transaction confirmation delay of the PBFT algorithm will greatly increase, and the throughput will be greatly reduced.

Variant PBFT-based consensus protocols have been proposed to solve the problem of poor scalability for PBFT [14]. Some works showed that the scalability of the consensus protocol can be well-improved when some certain technologies in the cryptograph field are combined. For example, PBFT with a short-lived signature was proposed in [18], where a short-length cryptographic key was used to sign or verify message in PBFT. A multi-signature scheme is also proposed to improve the scalability. It allows a group of signers to collaboratively sign a single message [19]. Multi-signatures, such as the Schnorr signature scheme [20] and Cosi multi-signature scheme [21], can significantly reduce the overall size of a block when allowing many signatures into the same block. FastBFT is presented in [22] with several optimizations, including a lightweight secret sharing scheme and hardware-based trusted execution environments. However, these methods [18,19,20,21,22] usually spend much time on encryption and decryption, and the security of the encryption algorithm needs to be further proven.

Another effective evolution path is sharding [23,24]. Each shard has its own consensus group where transactions can be packaged and committed within a relative short time. For instance, Delegated Byzantine Fault Tolerance (DBFT) proposed in [25] assumes that nodes are separated into a few clusters. It improves the scalability of the original PBFT [14] by reducing the number of nodes required to exchange confirmed information. However, it is proven that DBFT cannot bear even one Byzantine node in the cluster [26]. The security level of the shard-based protocol is low due to the lack of data sharing in different shards [27]. Similar to sharding, a multi-layer PBFT system was developed in [28] to reduce the cost of communication. A tree topology was constructed where each node was treated as a tree node. It can refrain the communication between nodes within their layers. However, it cannot be applied to a real blockchain, since it assumes that each subgroup contains nodes in the same amount, and faulty nodes only exist in the bottom layer [29].

In addition to grouping entire nodes, some methods have been proposed to improve the scalability by selecting partial nodes as the consensus group. For example, a consensus protocol based on reputation was proposed in [30], where reputation served as an incentive for good behavior. Nodes with a high reputation value are allowed to enter into the consensus group and commit transactions into the block. An optimized PBFT method based on the eigentrust model was proposed in [31]. It evaluates the trust of nodes through transactions between nodes and selects a certain number of nodes with high trust to participate in the consensus. However, the existing reputation-based consensus protocols do not carefully consider the node behavior in the process of transaction, and partial consensus violates the decentralized design principles for the blockchain.

In this paper, we propose a two-layer hierarchy structure, where nodes serve as different roles for the final consistency of the block. Nodes only need to exchange messages within their layers, therefore reducing the communication complexity and improving the scalability. To ensure the security of the network, a novel reputation model is designed to drive the Byzantine nodes to the bottom layer. The reputation values of the nodes are periodically updated according to the developed reward and punishment mechanisms. A random selection strategy has been developed to randomly select a leader node to balance the concentration of the leader node in high-reputation nodes. Some experimental results indicate that the proposed method has excellent performance in comparison to some state-of-the-art models.

To summarize, the main improvements of this paper are as follows:A hierarchy structure is designed to assign nodes to two layers. It is more scalable, since it reduces the communication complexity of the nodes to linear, compared to the square level of PBFT.A reputation model is proposed to evaluate the behaviors of nodes in the process of a transaction. It can be applied to disable malicious nodes from destroying the consensus and improve the security. The maximum proportion of the Byzantine nodes is higher than that in PBFT.A random selection strategy has been proposed to leverage the concentration of the leader node. Nodes with higher reputation values cannot get more of a chance of being the leader for them. Additionally, such a strategy can be applied to increase the robustness of a network by enhancing the unpredictability of the next leader, which is chosen in order in PBFT.

The remainder of this paper is organized as follows. A combination of local and global reputation model is described in Section 2. Some details of the HBFT consensus protocol are presented in Section 3. The performance analysis is presented in Section 4. The experimental results are discussed in Section 5, followed by some conclusions in Section 6.

## 2. Combination of Local and Global Reputation Models

Both normal and malicious nodes are included in the blockchain network. Some consensus protocols, such as PBFT [14] and DBFT [25], do not differentiate normal or malicious nodes before and after consensus. Reputation models have been proposed by some works [30,31] to distinguish normal nodes from malicious ones. Since the combination of local and global reputations was not considered in [30,31], it did not perform well in preventing malicious nodes from gaining high-reputation values.

Some details of the proposed reputation model are given to overcome the limitations mentioned above. Firstly, a local reputation model is described in Section 2.1, which is applied to evaluate every transaction and score the nodes with a local reputation value. A global reputation model is proposed in Section 2.2 to incorporate the local reputation values of different nodes by designed rules.

### 2.1. Local Reputation Model

In the transaction phase, a node may involve numbers of transactions with different nodes in the network. Let *t_ij_* be the transaction score for node *j* as scored by node *i*. For security concerns that malicious nodes may deliberately underscore a normal node, a mutual assessment mechanism has been proposed. Node *j* gives the same score to node *i* after it receives the scores. *T_max_* is set as the max waiting time for a node. During a transaction event, if a node does not receive the correct response within *T_max_*, the transaction event is regarded as a failure, and the penalty mechanism is triggered. Otherwise, the nodes in this transaction event would be rewarded.

In the case of a reward, a reward function for node *i* (from any other nodes) is:(1)$$rewards_{i}=\lambda_{1}*\left(R_{max}-R_{i}\right)*\frac{S_{count}}{h}*T_{resp}$$
(2)$$T_{resp}=1-\frac{T_{actual}}{T_{max}}$$
where *rewards_i_* is the positive scores that node *i* obtains from other nodes during a single transaction event. ∗ is a multiple operator. *λ*_1_ is a reward moderator to the magnitude of the increasement in the nodes’ reputation. *R_max_* denotes the maximum value of the reputation. *R_i_* is the current reputation of node *i*. *h* represents the number of historic rounds that are taken into account for node *i*, which will be discussed later. *S_count_* is set as the number of rounds that node *i* responses to properly in the past *h* rounds of transaction phases. *T_resp_* represents a time coefficient positively correlated with the node response time. *T_actual_* is the actual response time of the node.

The idea behind this reward mechanism is that if node *i* successfully completes many transactions during a series of continuous transaction phases, the reputation reward magnitude of node *i* will decrease gradually accordingly.

While a node refuses or delays the transaction request from other nodes in a required period, the node that trades with it will give a negative evaluation. If a node keeps failing to finish transactions, the reputation of the node will decrease reversely. The penalty value for node *i* is calculated as:(3)$$decay_{i}=-\lambda_{2}*\left(R_{i}-R_{min}\right)*\left(1-\frac{S_{count}}{h}\right)$$
where *decay_i_* is the negative scores that node *i* gets from the other node during a single transaction event. *λ*_2_ is a penalty moderator for the magnitude of decline in the nodes’ reputation. *R_min_* denotes the minimum value of the reputation.

Some previous works [30,31] tend to punish or reward nodes at a fixed value, which makes is difficult to differentiate normal and malicious nodes. In the proposed local reputation model, the reputation values of malicious nodes can be reduced in time, and the reputation values of normal nodes can be enhanced quickly. The scoring process in phase *n* is shown in Figure 1.

### 2.2. Global Reputation Model

The proposed local reputation model can improve the security by punishing malicious nodes and rewarding normal ones. However, malicious nodes may attack normal ones by underscoring them many times at one transaction stage. A global reputation model is designed to avoid such attacks by comprehensively considering the scores of all other nodes. Let *t_ij_* be the transaction score for node *i* evaluated by node *j* at a transaction event, which could be a reward or a decay, such as:(4)$$t_{ij}=\left\{ \begin{array}{ll} \sum_{r=1}^{N}t_{r}^{}/N, & N\geq 1\\ 0, & N=0 \end{array}\right.$$
where *t_r_* is the evaluation score of the *r*th transaction in *N* transactions. *N* is the number of transactions between *i* and *j*. The *N* transactions are independent from the evaluation events between nodes.
(5)$$c_{i}=\sum_{k}a_{k}t_{ik}=a_{1}t_{i1}+a_{2}t_{i2}+...+a_{ii-1}+a_{ii+1}...+a_{P}t_{iP}$$
(6)$$R_{i,n+1}=R_{i,n}+c_{i}$$
where *c_i_* is the final increment of the reputation value of node *i*. *P* is the number of nodes in the network. *a_k_* is a weight distribution function, which is used to weight the contribution of the transaction scores from other nodes towards node *i*. *a_k_* will be discussed later. *R_i,n_* is the reputation of node *i* at the transaction phase *n*.

To more intuitively show the relationship between the local and global reputation models, the update process of the reputation values is described in Algorithm 1 and Figure 2 as follows:
**Algorithm 1** Update node reputation value**Initialize:***R_max_* = 1, *R_min_* = 0, *λ*_1_ = 0.1, *λ*_2_ = 0.1, *T_max_* = 0.5 s
**Input:**
*R*, *h*, *T_actual_*, *P*, *N*
**for** 
j ∈ [0, P]
**then**
  **for** each
i ∈ [0, N]
**do**
 **if** 
$T_{\mathrm{actual}}\;\leq {\;T}_{\max}$
**then**
   
$T_{\mathrm{resp}}{=1\;-\;T}_{\mathrm{actual}}/T_{\max}$
  
${\mathrm{rewards}=\lambda }_{1}{*\;(R}_{\max}\;{-\;R)\;*T}_{\mathrm{resp}}{*S}_{\mathrm{count}}/h$
 **else**
  
${\mathrm{decay}=\lambda }_{2}{\;*\;(R\;-\;R}_{\min}{)\;*\;(1\;-\;S}_{\mathrm{count}}/h)$
 **end if**
  **end for**
   
$t_{j}=\sum_{r=1}^{N}t_{r}/N$
**end for**
 
$c=\sum_{j}a_{j}t_{j}$
R=R+c 
**Output: *R***

## 3. HBFT Consensus Protocol

### 3.1. Hierarchy Structure

The data throughput of PBFT [14] is low due to the high communication complexity between nodes. A tree topology structure was proposed in [28,29] to reduce communication complexity by assigning nodes to several different layers. It can be applied to improve the data throughput of a network. However, it is difficult to ensure the consistency with malicious nodes existing in the high layer.

A two-layer hierarchical structure based on node reputation is designed to address the problem mentioned above. Each node is assigned into one of the two layers according to its current reputation value. Nodes in the network are ranked by reputation values from high to low after the evaluation of the transaction stage. Some nodes in the top ranking of the reputation value are placed in the high layer, and the rest are placed in the low layer. In this way, both the node scalability and data throughput of the network can be improved.

Suppose there are *H* nodes in the high layer, then the nodes in the low layer are accordingly separated into *H*-1 groups, called subgroups. In the two-layer structure, there are clients, consensus nodes and subleader and leader nodes. The leader node is responsible for the collection of the transaction data and packages for blocks in the entire blockchain. The subleader node is the candidate of the leader node. It can participate in the verification of blocks and broadcast blocks to the consensus nod in its subgroup. The consensus node is responsible for verifying the blocks received from the subleader node. It can check and report malicious behavior by a subleader node. The client can initiate transactions by connecting to one of nodes in the blockchain. The overall hierarchy structure is shown in Figure 3.

The number of nodes in each subgroup is not fixed and affects the final performance of the network. The distribution strategy of the nodes in the low layer will be discussed later.

The block data transmission process of HBFT is described in Figure 4.

The main processes are as follows:
(1)Request phase. The client initiates a transaction request to the node it connects to. It signs the transaction with private key. The request format is:
(7)$$<<request,t,d,g(d),c,id>,Sig_{c}>$$
where *t* is the timestamp. *d* is the transaction data. *g*(*d*) is a hash value of the transaction data *d*, *c* is a client identification, *id* is an identification number of the leader node and *Sig_c_* is the client signature. The node verifies the identity of the client and the timestamp *t* in the blockchain. If the authentication is successful and the timestamp is not out of date, the node sends it to the leader node.(2)Prepare phase. After receiving the request message from the client, the leader node will order and package the transaction data into a block and then broadcast it to the subleader node. The format of the message is:
(8)$$<<prepare,t,d,g(d),id>,m,Sig_{leader}>$$
where *m* is request message, and *Sig_leader_* is the signature of the leader node.
(3)Lpre-prepare phase. The subleader node confirms whether the signature is correct. If the verification is successful, it signs with *Sig_sub_* and forwards the message to the nodes in its subgroup. The format of the message is:
(9)$$<<lpre-prepare,t,d,g(d),id>,m,Sig_{sub}+Sig_{leader}>$$
(4)Lprepare phase. After the consensus node in the subgroup verifies the signatures of both the leader node and subleader node, it will send a confirmation message signed with *Sig_i_* to the other nodes in the same subgroup, called the lprepare message. The format of the message is:
(10)$$<<lprepare,t,d,g(d),id>,m,Sig_{i}>$$
(5)Lcommit phase. When a consensus node collects 2*f*+1 correct lprepare messages, it will send a commit message to the subleader node. The format of the message is:
(11)$$<<lcommit,t,d,g(d),id>,m,Sig_{i}>$$
(6)Reply phase. If the subleader node receives more than half-valid messages in the lcommit phase, it commits the block to the blockchain and replies to the client. The format of the message is:
(12)<<reply,t,d,g(d),id>,m>


### 3.2. Leader Selection Mechanism

The leader node is responsible for the block packaging and distribution. The state of the leader node determines the security and efficiency of the consensus protocol. The leader node in [10,11] was usually selected as the one who occupied the most computation power or stakes. However, it is energy-consuming and may cause an insecure concentration of the computation power or stakes. A consistent and trust fusion method was proposed in [29], where a node with a higher reputation is more likely to be selected as a leader. It does not consume much energy; however, it harms the fairness of the blockchain and may cause an insecure concentration of the reputation. A random selection consensus protocol was presented in [32]. It adopted a verifiable random function to select a committee that includes a leader node as well as a set of verifier nodes. It guarantees the randomness of the selection process. However, it requires the frequent replacement of committee members, which is inefficient.

To overcome the limitations mentioned above, a random selection mechanism is designed. Every node is qualified to participate in the consensus process. In order to enable a node to prove that it is selected as the leader, the mechanism requires node *i* to have a public/private key pair (*pki* and *ski*), and the nodes in the network do not keep any private state, except for their private keys. The mechanism is implemented using verifiable functions (VRFs) [33]. For any legal input *x*, VRF*_sk_*(*x*) returns two values: a hash and a proof. The hash is a hashlen-bit-long value that is only determined by *sk* and *x*. For any node that does not know *sk*, it is random and indistinguishable. The proof enables the nodes to verify that the hash truly corresponds to *x* without knowing *sk*.

At the beginning of each consensus epoch, there is a short phase that the nodes need to calculate with the given seeds using VRF and exchange messages for their computation results. The minimum one is selected as leader of this epoch as the one titled with leader in Figure 5. The items in the shared data are publicly known by every node. The randomness of the proposed selection mechanism comes from a publicly known seed. The processes of seed generation and distribution are shown in Figure 5.

The seeds should be public and cannot be controlled by the attacker. For each epoch of consensus, a new seed is generated. The seed of epoch *r* is determined by the current leader using VRFs with seeds in the previous epoch. *seed_r_* and proof *cert* can be calculated as follows:(13)$$seed_{r}=VRF\_hash(sk_{i},seed_{r-1})$$
(14)$$cert=VRF\_proof(sk_{i},seed_{r-1})$$

The value of *seed*_0_ can be chosen randomly using distributed random number generation [34]. The seed and corresponding proof are additionally added into the proposed block. As long as the block of epoch *r* reaches a consensus, every node knows the seed for the next epoch. The block broadcast process has been illustrated in Figure 4. The node can verify that *seed_r_* is indeed produced from *seed_r−1_* by the leader with the leader’s *pk* and *cert*.
(15)$$seed_{r}=VRF\_P2H(cert)$$
(16)$$True/False=VRF\_verify(pki,seed_{r-1},cert)$$

The leader selection mechanism is triggered by the following conditions. Firstly, the leader node has its own term of office called the epoch. An epoch usually includes multiple rounds of the consensus. Once the epoch of the leader node is reached, the nodes in the consensus group will automatically select another leader. This can improve the decentralization of the network, as each node has the opportunity to be the leader node in a limited time. Secondly, the leader node may crash unexpectedly due to a network delay or other reasons. In order to keep the network working, the epoch of the current leader node is killed, and another leader is selected. The proposed random selection mechanism can ensure the fairness and security of the network by selecting the leader node randomly.

## 4. Performance Analysis

### 4.1. Communication Complexity

Communication complexity is an important index of the consensus protocols. It can be reflected by the number of communications times. The communication complexity of PBFT [14] is O(*P*^2^). It means the number of communication times increases exponentially with *P*. In our HBFT, the communication complexity is optimized.

Suppose there are *l* nodes in each subgroup and the number of nodes in each subgroup is consistent. The total number of nodes *P* in the network is:(17)P=1+(H−1)×l

For the nodes in the low layer, the number of communication times *C*1 is:(18)$$C1=(H-1)\left[{\left(l+1\right)}^{2}+2l\right]$$

For the nodes in the high layer, the number of communication times *C*2 is:(19)C2=2H

Therefore, the gross number of communication times *C* in the network is:(20)$$C=C1+C2=(H-1)\left[{\left(l+1\right)}^{2}+2l\right]+2H$$

*C* can also be expressed in *P* according to Equation (17) as:(21)$$C=\frac{l^{2}+4l+3}{l}(P-1)+2$$

One can find from Equation (18) that the gross number of communication times *C* increases linearly with the total number of nodes *P*. In other words, the developed consensus protocol can significantly reduce the communication complexity by comparison with PBFT [14]. It will be proven in the experiments later.

### 4.2. Byzantine Fault Tolerance

The Byzantine fault tolerance reflects the security of the consensus protocol. The more Byzantine nodes a consensus protocol can tolerate, the securer it is. The upper limit of the Byzantine node in PBFT [14] is (P − 1)/3. In our HBFT, the Byzantine fault tolerance rate is optimized. In addition to the leader node, the total number of nodes in the high layer *x* is:(22)$$x=\frac{P-1}{l}$$
where *l* ≥ 3.

Since there is no Byzantine node in the high layer, a consensus can be reached as long as more than half of the subleader nodes reply correctly. During the consensus process in the low layer, the subleader node in each subgroup needs to receive more than one-third of the correct messages. In the worst case, half of subgroups are comprised of Byzantine nodes, and the subleader node cannot reply correctly. The maximum number of Byzantine nodes in HBFT is:(23)$$\frac{x}{2}*\frac{l}{3}+\frac{x}{2}*(l-1)=\frac{P-1}{3}*\frac{2l-2}{l}$$

Since *l* ≥ i3, we can conclude that the developed consensus protocol can tolerate more Byzantine nodes.

## 5. Experiments

In order to test the effectiveness of the proposed protocol, an experimental network was built as follows. Every node is running on a physical or virtual machine with equivalent performances. We initiate nodes with different data transmission delays. Every node can participate in the consensus, and no one would be excluded from the network. Each node keeps two ledgers: one for recording the reputation value in the transaction phase and the other for recording the transactions in the consensus process. For security experiments, we set some nodes as malicious ones. They can hide or tamper transaction data, deliberately underscore normal nodes and even collaborate with their partners to get high scores. The normal node always responded correctly in time.

### 5.1. Reputation Model Parameters

Reputation is an important index for assigning nodes in the two-layer structure. To improve the security of network, some parameters should be optimized for two purposes: one is to disable malicious nodes from getting high scores, and the other one is to reduce the negative influence that malicious nodes bring to normal nodes. In this experiment, *R_max_* was set to 1, and *R_min_* was set to 0. Since the experiment was simulated in a local area network, *T_max_* was set to 1000.

#### 5.1.1. Historic Rounds

When a node scores for its transaction node, historic rounds are helpful in judging whether the node is malicious. In the experiment, we simulate three normal nodes (node0, node1 and node3) and a malicious node (node2). In order to get the optimal number of *h*, *h* is changed from 1 to 9 at an interval of 2. Additionally, we set the proportions of the transactions between three normal nodes and the malicious node with different values, which are 6.25% for node0, 62.5% for node1 and 31.25% for node3, respectively. Some results are shown in Figure 6, respectively.

It can be seen from Figure 5 that the number of *h* has a significant influence on the reputation values of the nodes. The malicious node is successfully blocked from getting high reputation values, but the normal nodes are influenced by the malicious node to different extents with the changes of the *h* values. When *h* is not considered (*h* = 0), the gap of the reputation value between normal nodes is expanding. It indicates that the local reputation model without *h* cannot resist attacks from the malicious node well.

We selected standard deviation to quantitatively evaluate the negative influence of the malicious node towards the normal nodes. We selected reputation values of the normal nodes at 200 rounds and computed the standard deviation. The results are given in Table 1. The lower standard deviation is, the better the performance of the reputation model.

One can find from Table 1 that, when *h* is set to 3, the standard deviation is the minimum. In the subsequent experiments, *h* is set as 3 and kept the same.

#### 5.1.2. Weigh Selection Distribution

The malicious nodes may expand impact of their own evaluations by fictionalizing their reputation values. In a proposed global reputation model, *a_k_* is introduced to assign each node with different weights. It is only related to the reputation values of the nodes. The value of *a_k_* is between 0 and 1. The sum of *a_k_* is 1.

Two kinds of weight selection functions will be discussed. The default one is the uniform distribution, and the other one is the linear distribution. For uniform distribution, the weights are all equal:(24)$$a_{k}\equiv 1/(P-1)$$

For linear distribution, suppose *k* is the slope. The reputation values of each node are listed in descending order and mapped into certain equidistant values in [0, *b*], and *b* can be calculated as follows:(25)$$b=\frac{2}{k(P+1)}$$

In the experiment, we simulated a group of nodes, including multiple malicious nodes. We set *f* as the proportion of the malicious nodes in the network. Since the maximum value of *f* in PBFT [16] does not exceed one-third, *f* is set to one-sixth, 1/one-fourth and one-third, respectively. Some simulation results are shown in Figure 7.

One can find from Figure 7a–c that the gap of the reputation values between the normal and malicious nodes is narrowed with the increase of *f*. When the distribution is uniform and *f* is one-third, the reputation value of the malicious nodes is almost the same as that of the normal nodes after 2000 rounds. It denotes that the node with a higher reputation could be malicious.

We set the distribution as linear and *f* as one-third, and the results are shown in Figure 7d. The reputation value of normal nodes apparently surpasses that of malicious nodes at around 800 rounds of transaction phase. It shows that, when the distribution is linear, the malicious nodes can be well-disabled. The distribution is set as linear and remains unchanged in the later experiment.

### 5.2. Node Allocation Scheme

The existing research has provided many solutions for node management. For example, a smart collaborative balancing scheme was proposed in [35] to dynamically adjust the orchestration of the network and smartly allocate the bandwidth for each node. A node overhaul scheme proposed in [36] could efficiently improve the network lifetime by creating a uniform cluster with good quality. It mainly considers the size of the cluster and total intra-cluster communication distance.

In the proposed protocol, the nodes are allocated into two layers. Consensus nodes are required to be assigned to some subgroups. *w* is set as the ratio of consensus nodes to subleader nodes. Three different allocation situations are simply discussed, including average, random and geographic, respectively. In an average situation, each subgroup has the same number of nodes. The number of nodes in each subgroup is uncertain in a random situation. It may be 0 or infinite. In the geographic situation, the real position of the nodes is taken into consideration, which denotes that the adjacent nodes are placed in the same subgroup.

Data throughput and latency are two important indexes of the consensus protocols. Data throughput is expressed as the number of transactions per unit time (TPS):(26)$$TPS=\frac{transactions}{\Delta t}$$

The higher data throughput denotes that the consensus protocol is more efficient. Latency represents the time difference from transaction submission to transaction confirmation. The lower latency denotes a better performance of the consensus protocols. In the experiment, the value of *w* is changed from 1 to 19 at an interval of 2. The total number of nodes in the network is fixed at 61. Meanwhile, the consensus nodes are allocated into different subgroups in three ways, as mentioned above. Some results are shown in Figure 8.

It can be seen from Figure 8 that the random scheme performs the worst in both data throughput and latency. In the average and geographic schemes, as the *w* increases, the data throughput decreases, latency increases. Both data throughput and latency are optimal when *w* is 1 for the discussed schemes. The performance of the geographic scheme is the best among the investigated ones. The geographic scheme is taken as the node allocation one, and *w* is set as 1 in the subsequent experiment.

### 5.3. Comparisons with Relevant Consensus Protocols

In order to further evaluate whether the proposed protocol is efficient, the proposed protocol was compared with PBFT [14] and T-PBFT [31] in communication times, data throughput and latency. The less the communication times, the higher the node scalability of a blockchain is. The number of transactions is set as 3000. The number of nodes is changed from 4 to 40 at an interval of 3. Some results are given in Figure 9.

One can find from Figure 9 that the performance of our developed HBFT is the best for the communication times, data throughput and latency among the investigated protocols [14,31]. Some reasons are as follows: the multiple confirmation phase is introduced in [14] to ensure the consistency and correctness of the final block in the presence of malicious node. However, it would greatly increase the communication times. The eigentrust model is utilized to reduce the communication times in [31]. However, it does not basically solve the problem by only allowing nodes with a high reputation to participate in the consensus process. It would cause centralization of the network, which violate the guideline of blockchain. Our developed protocol can be applied to reduce the communication times by the hierarchy structure. The proposed hierarchy structure makes nodes only need to exchange messages with their related nodes. The reputation model is utilized to assign some reliable node to the high layer, where the confirmation phase is shortened. It effectively increases the data throughput and reduces the latency and proposed leader selection mechanism, which can ensure the security of the network by the fairly selected leader node.

In order to get fairer results, some nonquantitative indicators are taken into account. The results are based on the best parameters provided by the authors in their protocols [12,14,25,31]. Some comparison results are shown in Table 2.

It can be found from Table 2 that the proposed HBFT has an excellent performance among the investigated protocols [12,14,25,31]. Some reasons are as follows: DPOS [12] cannot tolerate the Byzantine faults due to the lack of multiple validation during block generation. PBFT [14] is inefficient in the network composed of large-scale nodes with the communication complexity of O(*P*^2^). DBFT [25] and T-PBFT [31] reduce the degree of de-centralization to some extent. Our proposed protocol can reduce the communication complexity to O(*P*) with a high degree of decentralization. Additionally, the developed protocol selects the leader node randomly and fairly.

## 6. Conclusions

A novel consensus protocol HBFT based on the node reputation was proposed. A hierarchy structure has been developed to separate nodes into two layers. It deducts the communication times between nodes and well improves the scalability. A combination of local and global reputation models has been proposed to evaluate the behaviors of nodes in the network. Malicious nodes are disabled from getting into the high layer, which enhances the security in the network and speeds up the consensus in the high layer. Additionally, a random selection mechanism was proposed to ensure the fairness of the leader node. Some experimental results highlight that the proposed consensus protocol has an excellent performance in comparison to some state-of-the-art models. Compared with PBFT [14] and T-PBFT [31], the proposed protocol shows a better performance in low communication complexity, low latency and high throughput. Additionally, it can tolerate more Byzantine nodes and maintain high degrees of decentralization. For future works, we will continue to increase the number of layers in this hierarchy structure and develop a more effective reputation model that can disable malicious nodes in a high proportion.

## Figures and Tables

**Figure 1 sensors-22-05887-f001:**
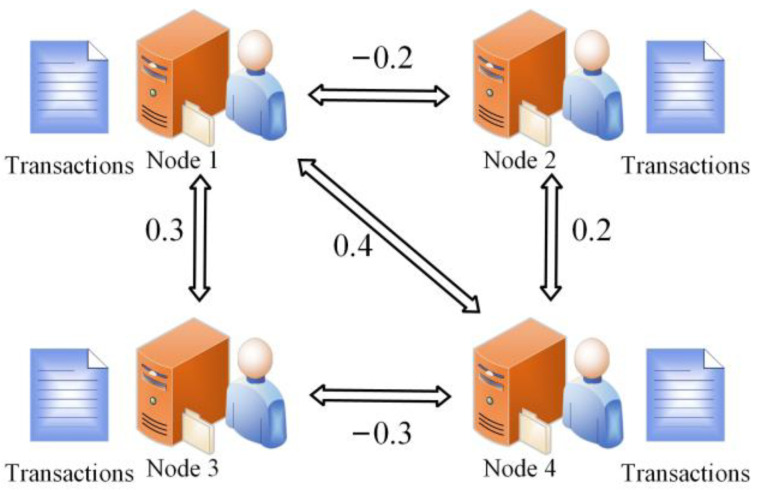
Transaction scores in phase *n*.

**Figure 2 sensors-22-05887-f002:**
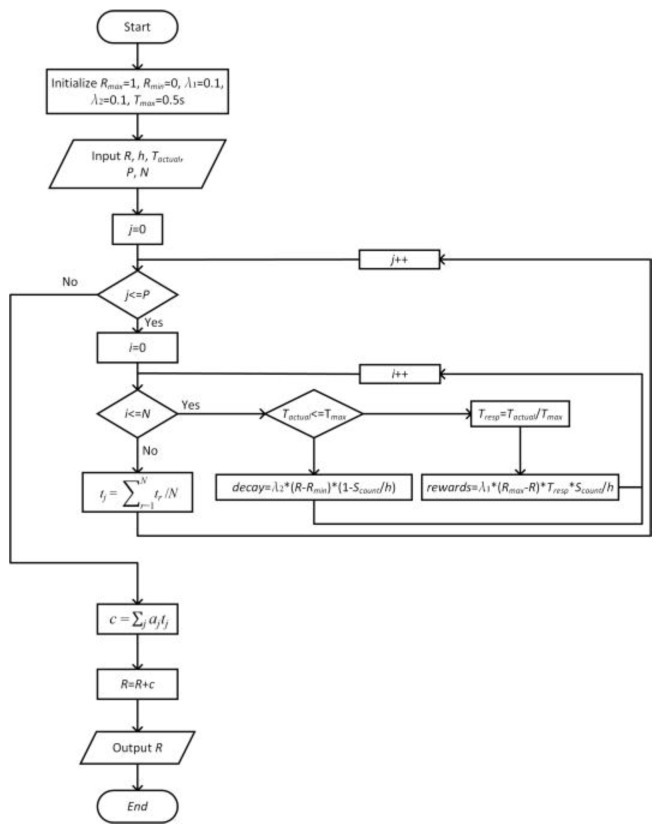
The flowchart of reputation updating process.

**Figure 3 sensors-22-05887-f003:**
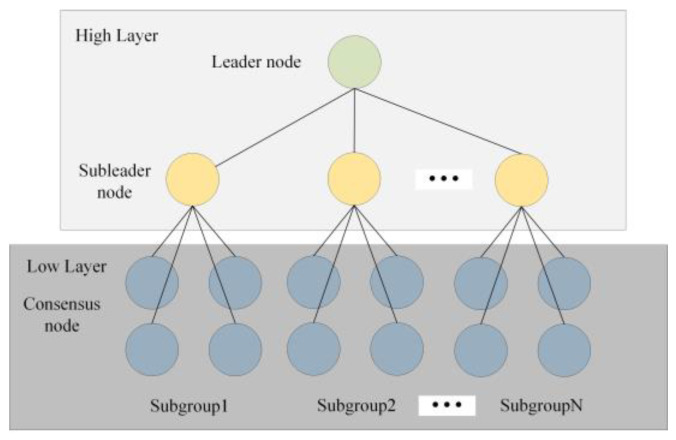
Hierarchy structure in HBFT.

**Figure 4 sensors-22-05887-f004:**
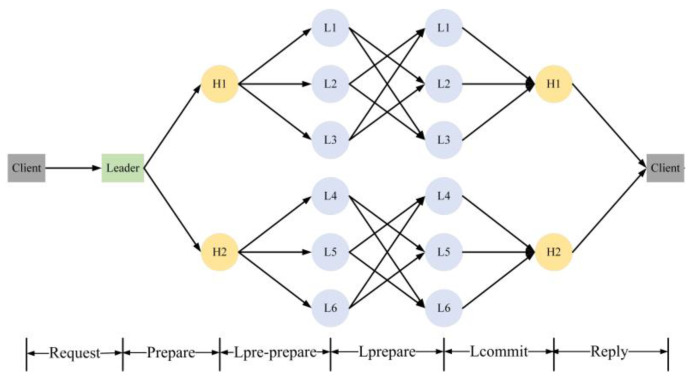
The overall flow diagram of HBFT.

**Figure 5 sensors-22-05887-f005:**
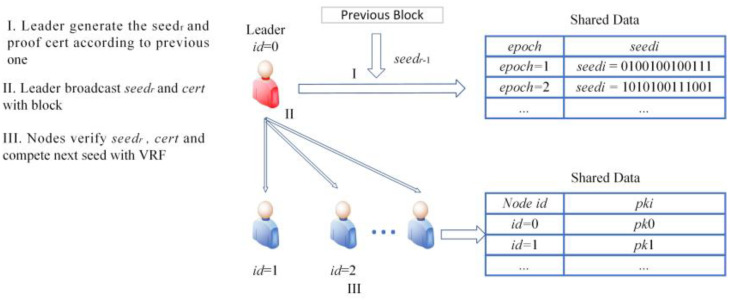
Leader selection mechanism.

**Figure 6 sensors-22-05887-f006:**
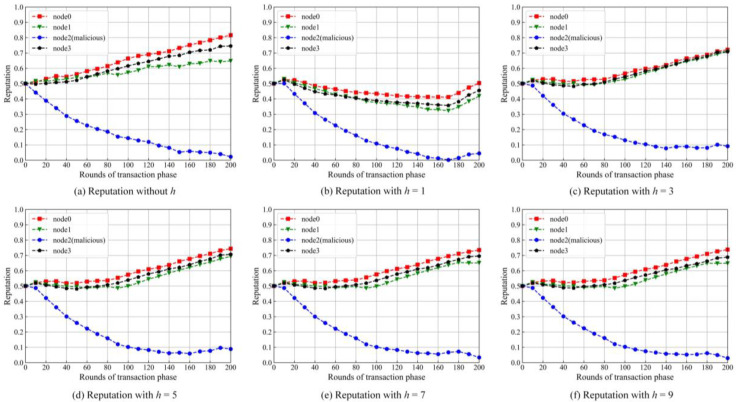
Trend of reputations with different *h*s.

**Figure 7 sensors-22-05887-f007:**
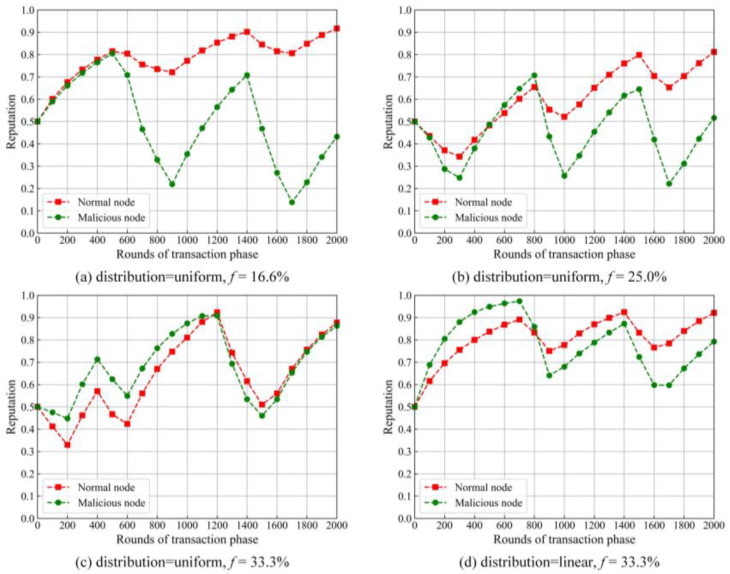
Trend of reputations with different *f*s and distributions.

**Figure 8 sensors-22-05887-f008:**
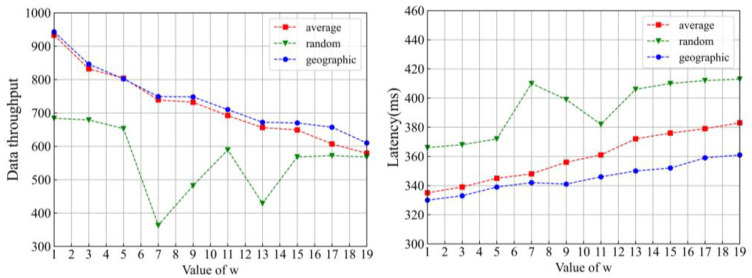
Data throughput and latency with different node allocation schemes in different *w*s.

**Figure 9 sensors-22-05887-f009:**
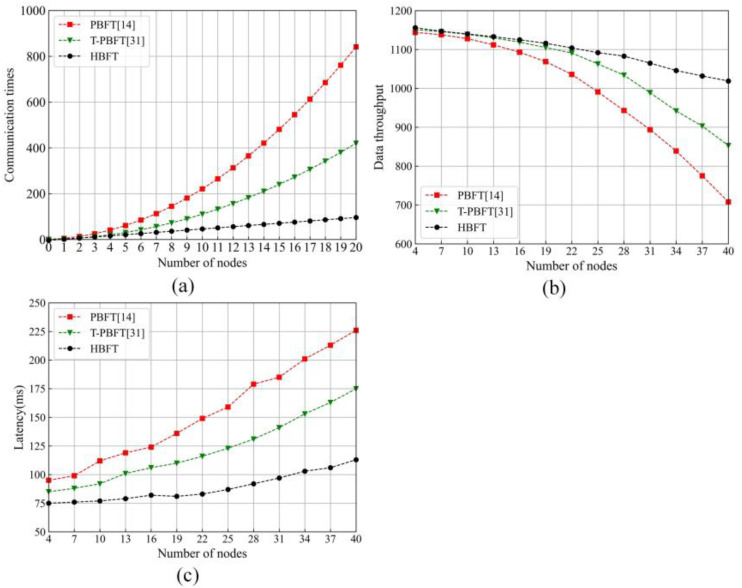
**Performance comparisons.** (**a**) is comparison of the communication times, (**b**) is comparison of data throughput, and (**c**) is comparison of latency.

**Table 1 sensors-22-05887-t001:** Standard deviation between normal nodes with different *h*s.

*h*	Standard Deviation
0	0.0843
1	0.0421
**3**	**0.0069**
5	0.0254
7	0.0420
9	0.0446 ^i^

^i^ The optimal *h* value refer to the minimum of standard deviation.

**Table 2 sensors-22-05887-t002:** Comparisons with different consensus protocols.

	PBFT [14]	DBFT [25]	T-PBFT [31]	DPOS [12]	HBFT
Communication complexity	O(*P*^2^)	O(*P*^2^)	O(*P*^2^)	O(*P*)	**O(*P*)**
Energy saving	Yes	Yes	Yes	Yes	**Yes**
Byzantine fault tolerance	(P -1)/3	(P - 1)/3	(2P - 1)/3	No	(P - 1)/3∗(2l - 2)/l
Scalability	Low	High	High	High	**High**
Degree of de-centralization	High	Medium	Medium	High	**High**
Fairness of Leader node	Medium	Medium	Low	Low	**Medium**

Some non-quantity comparisons are made.

## Data Availability

Not applicable.

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
