# Peer review of "A Hierarchy Byzantine Fault Tolerance Consensus Protocol Based on Node Reputation"

_sensors, 2022, doi:10.3390/s22155887_

Round 1
Reviewer 1 Report
The current version looks good.
Author Response
Thank you very much for your affirmations on our manuscript.
Reviewer 2 Report
This paper presents an interesting work to improve the the scalability, efficiency and security of consensus protocol of a blockchain with a reputation model with a rewarding mechanism.
The paper is good and scientifically sound.
The introduction is nice and provide a good context with important literature references. Nevertheless it should better point out in a paragraph what are the main improvements of this approach over similar ones.
The quality of the artworks of figure 2 and 4 should be improved.
In section 3, besides the pseudocode, a flowchart of the could help the reading of their proposed algorithm.
Section 4 and 5 are quite interesting and well motivated.
Section 6, should provide some information about the results obtained to summarize the finding of the research
Author Response
We are grateful to the anonymous reviewer for your valuable comments and suggestions which help improve the quality of our manuscript.

This manuscript is a resubmission of an earlier submission. The following is a list of the peer review reports and author responses from that submission.
Round 1
Reviewer 1 Report
Review of paper 1673068
Researching A Hierarchy Byzantine Fault Tolerance Consensus Protocol Based on Node Reputation, nowadays, when networks and node’s reliability are very important aspects in shaping the protocol for artificial societies, it is interesting and challenging for the authors. Submitted paper defines a novel hierarchy structure for the o improvement of the scalability by assigning nodes to a high layer or a low layer according to their reputation values. There are at least three lines which are involved in this case, such as the following: public chain, consortium chain, and private chain. First one of these is a network that arbitrary nodes are allowed to participate in at any time. There is no apparent factor of overall control and legal compliance oversee, like there is in the private chain, where nodes are all controlled by one institution. The conclusions and observations of experimental part show that proposed consensus protocol might have practical result, which must be further researched.
Introduction | |
Does the introduction provide sufficient background information for readers not in the immediate field to understand the problem/hypotheses? | Yes. The first chapters of the paper make the presentation for one of the latest trends in distributed networks: Blockchain, which consists of public chain, consortium chain, and private chain. |
Are the reasons for performing the study clearly defined? | Yes. In my opinion the reasons for developing the research seem quite clear. |
Are the study objectives clearly defined? | The objectives must be clearly defined in the Introduction. |
2. Literature Review and Model Development | |
Is the literature cited balanced or are there important studies not cited, or other studies disproportionately cited? | The literature referred in this paper is supporting to the research. |
Please identify statements that are missing any citations, or that have an insufficient number of citations, given the strength of the claim made. | - |
3. Methodology and Data | |
Are the methodology and data used appropriate to the purpose of the research? | Yes, but it may be further developed with more experimental tests. |
Is sufficient information provided for a capable researcher to reproduce the experiments described? | The authors should probably provide more information about the blockchain interactions and its outcome. |
Are any additional experiments required to validate the results of those that were performed? | Other experiments may improve the definition level of the blockchain network. |
Are there any additional experiments that would greatly improve the quality of this paper? | Yes. |
Are appropriate references cited where previously established methods are used? | Yes |
4. Results | |
Are the results clearly explained and presented in an appropriate format? | May be improved |
Do the figures and tables show essential data or are there any that could easily be summarized in the text? | May be improved |
Are any of the data duplicated in the graphics and/or text? | No. |
Are the figures and tables easy to interpret? | It may be improved. |
Are there any additional graphics that would add clarity to the text? | Yes. |
Have appropriate statistical methods been used to test the significance of the results? | Yes. |
5. Conclusions and Implications | |
Are all possible interpretations of the data considered or are there alternative hypotheses that are consistent with the available data? | May be further improved. |
Are the findings properly described in the context of the published literature? | Yes. |
Are the limitations of the study discussed? If not, what are the major limitations that should be discussed? | May be further developed. |
Are the conclusions of the study supported by appropriate evidence or are the claims exaggerated? | Conclusions may be further developed. |
Significance and Novelty | |
Are the claims in the paper sufficiently novel to warrant publication? | Yes. |
Does the study represent a conceptual advance over previously published work? | Yes. |
Journal Selection | |
Is the target journal (if known) appropriate? If not, why not? | Yes |
What is the likely target audience of this paper? | Network engineers. Artificial societies. Blockchain data. |
Minor comments | |
Please refer to the comments in the edited manuscript file for minor comments. | |
Major comments | |
To publish this paper in your target journal, the following revisions are strongly advised: | Conclusions must be provided with statistical evaluation of the results and findings. |

Author Response
We really appreciate your comments. It is very helpful for us. We have carefully revised our manuscript according to your comments. Thank you again!

Reviewer 2 Report
In this paper, a reputation model is proposed to distinguish normal nodes from malicious ones by updating their reputation values. All nodes in the blockchain network are involved in a mutual transaction phase. During the phase nodes reward or punish their partners, update their reputation values and broadcast it. A hierarchy structure is designed to improve the scalability by assigning nodes to a high layer or a low layer according to their reputation values.
- The motivation of this paper should be clearly stated in abstract and introduction part. Why Hierarchy Byzantine Fault Tolerance Consensus Protocol is critical for current network?
- Figure 2 is too simple (only two layers). More details should be added to illustrate the hierarchy structure.
- “After calculation, the leader node broadcasts L and proof to other nodes in the network.” Any security considerations for that?
- “The value of id is totally random” how to ensure that?
- Some latest progresses in node management, such as Smart Collaborative Balancing for Dependable Network Components in Cyber-Physical Systems, A Node Overhaul Scheme for Energy Efficient Clustering in Wireless Sensor Networks, should be added and compared.
- The errors and typos should be rechecked.
Author Response
We really appreciate for your comments. They are very helpful to us. We have carefully revised our manuscript according to your comments. Thank you again!

Round 2
Reviewer 2 Report
The authors didn't check the previous comments carefully.